# Cell Cycle Regulation of Hippocampal Progenitor Cells in Experimental Models of Depression and after Treatment with Fluoxetine

**DOI:** 10.3390/ijms222111798

**Published:** 2021-10-30

**Authors:** Patrícia Patrício, António Mateus-Pinheiro, Ana Rita Machado-Santos, Nuno Dinis Alves, Joana Sofia Correia, Mónica Morais, João Miguel Bessa, Ana João Rodrigues, Nuno Sousa, Luísa Pinto

**Affiliations:** 1Life and Health Sciences Research Institute (ICVS), School of Medicine, University of Minho, Campus Gualtar, 4710-057 Braga, Portugal; apinheiro@med.uminho.pt (A.M.-P.); anaritasantus@gmail.com (A.R.M.-S.); nda2114@cumc.columbia.edu (N.D.A.); id8212@alunos.uminho.pt (J.S.C.); monica.dias-morais@pasteur.fr (M.M.); joaobessa@med.uminho.pt (J.M.B.); ajrodrigues@med.uminho.pt (A.J.R.); njcsousa@med.uminho.pt (N.S.); 2ICVS/3B’s—PT Government Associate Laboratory, 4805-017 Braga/Guimarães, Portugal; 3B’nML—Behavioral &Molecular Lab, 4715-057 Braga, Portugal; 4Department of Psychiatry, New York State Psychiatric Institute, Columbia University, New York, NY 10032, USA

**Keywords:** unpredictable chronic mild stress, depression, antidepressants, hippocampal cytogenesis, cell cycle

## Abstract

Changes in adult hippocampal cell proliferation and genesis have been largely implicated in depression and antidepressant action, though surprisingly, the underlying cell cycle mechanisms are largely undisclosed. Using both an *in vivo* unpredictable chronic mild stress (uCMS) rat model of depression and *in vitro* rat hippocampal-derived neurosphere culture approaches, we aimed to unravel the cell cycle mechanisms regulating hippocampal cell proliferation and genesis in depression and after antidepressant treatment. We show that the hippocampal dentate gyrus (hDG) of uCMS animals have less proliferating cells and a decreased proportion of cells in the G2/M phase, suggesting a G1 phase arrest; this is accompanied by decreased levels of cyclin D1, E, and A expression. Chronic fluoxetine treatment reversed the G1 phase arrest and promoted an up-regulation of cyclin E. *In vitro*, dexamethasone (DEX) decreased cell proliferation, whereas the administration of serotonin (5-HT) reversed it. DEX also induced a G1-phase arrest and decreased cyclin D1 and D2 expression levels while increasing p27. Additionally, 5-HT treatment could partly reverse the G1-phase arrest and restored cyclin D1 expression. We suggest that the anti-proliferative actions of chronic stress in the hDG result from a glucocorticoid-mediated G1-phase arrest in the progenitor cells that is partly mediated by decreased cyclin D1 expression which may be overcome by antidepressant treatment.

## 1. Introduction

Most mammals, including humans, retain the ability to remodel the brain by incorporating new neurons in the adult hippocampal dentate gyrus (hDG) in a multistep process called adult neurogenesis [1,2]. Although recent conflicting reports have reinstated the scientific discussion about whether new neurons are generated in the adult mammalian hippocampus, compelling key evidence that has been gathered over the past 20 years strongly supports the idea that adult neurogenesis exists in humans [3,4,5,6,7,8].

Converging data reveal that these new neurons are critical for spatial memory, cognitive flexibility, and pattern separation but that they are also critical for mood and emotion [9,10,11,12,13]. Given this involvement in complex behaviors, it is plausible that neurogenesis disruption impacts neuronal circuitry and may ultimately lead to the development of disorders. Indeed, one of the most prominent observations in modern psychiatry is that hippocampal neurogenesis is decreased in animal models of depression and depressed patients and is re-established after antidepressant (AD) therapies [14,15,16,17]. We and others have shown that newborn cells at distinct stages of maturation contribute distinctively to memory as well as depressive and anxiety-like behavior [15,18,19,20,21,22]. However, the underlying mechanisms regulating this phenomenon and the molecular cues guiding its course are far from being completely understood. 

The generation of new cells from a neural progenitor involves several steps, namely cell division, cell fate commitment, and migration and maturation as well as the establishment of proper synaptic connections that culminate with a full integration of cells in the pre-existing network [23,24]. As in other tissues, the cell cycle of adult hippocampal progenitor cells is governed by cyclins and cyclin-dependent kinases (Cdk) that promote cell cycle progression, whereas Cdk inhibitors restrain it, stimulating cell cycle exit and post-mitotic state maintenance. Still, the cell cycle changes implicated in the decreased cell genesis occurring in the hippocampus of depressed individuals and animal models of depression and the pro-neurogenic/gliogenic effects produced by ADs are poorly comprehended. Whereas previous works have shown the implication of Cdk inhibitors in the decreased proliferative capacity of progenitor cells in the subgranular zone (SGZ) of the hDG in animal models of depression [25,26,27], no studies have looked at this phenomenon using an integrated multimodal approach.

To understand the cell cycle mechanisms regulating hippocampal cell proliferation and cell genesis in the context of depression and ADs treatment, we used two complementary approaches: an *in vivo* multidimensional rat model of depression—unpredictable chronic mild stress (uCMS), and an *in vitro* tool—rat hippocampal-derived neurosphere culture. Chronic stress exposure is a major precipitating factor of depression. Both in humans and animal models, chronic stress exposure may induce a disruption of the HPA axis, resulting in increased glucocorticoids (GCs) secretion, which ultimately leads to loss of corticosteroid feedback inhibition. Glucocorticoid receptors (GRs) are largely expressed in the hippocampus, thus rendering this region particularly susceptible to the effects of stress. Excessive GCs increase Ca^2+^ influx, making the neurons more vulnerable to excitotoxicity and oxidative stress [28,29,30]. These molecules have potent growth-inhibitory effects in several peripheral tissues and in the hippocampal region, thus inhibiting proliferation and neurogenesis [31,32,33,34]. In this study, we characterized the cell cycle of hippocampal progenitor cells in depressive-like and fluoxetine-treated animals. To mimic this elevation of GCs after chronic stress exposure in an *in vitro* setting, we stimulated rat hippocampal neurosphere cultures with dexamethasone (DEX), an agonist of GR. Cells were then treated with serotonin (5-HT), one of the main neurotransmitters implicated in fluoxetine treatment. Using this multidimensional approach, we could further elucidate the cell cycle-related molecular players and mechanisms regulating proliferation in the hippocampal neurogenic niche in the context of depression and AD treatment.

## 2. Materials and Methods

### 2.1. Animals and Treatments

Male Wistar rats (200–250 g, 2-month-old; Charles River Laboratories) were maintained under standard laboratory conditions (lights on: 08:00–20:00 h; 22 °C, relative humidity of 55%, ad libitum access to food and water). Rats were randomly assigned to one of the following groups (*n* = 10/group): non-stressed control vehicle (CT); stress-exposed (uCMS) vehicle; and uCMS fluoxetine (uCMS + FLX). A validated uCMS protocol was applied for 6 weeks (Appendix A), as previously described [20]. During the last 2 weeks of the uCMS, animals were injected intraperitoneally daily with fluoxetine (10 mg/kg fluoxetine hydrochloride in ultra-pure water; CAS No. 59333-67-4, Kemprotec, Middlesborough, UK) or saline, respectively. The dose was chosen based on previous studies [15,20,35,36]. All procedures were conducted in accordance with EU Directive 2010/63/EU guidelines on animal care and experimentation and were approved by the University of Minho Subcommittee of Ethics for the Life and Health Sciences (SECVS068/2017). 

### 2.2. Behavioral Analysis

All animals included in this study (*n* = 10) were assessed for behavior.

#### 2.2.1. Sucrose Consumption Test (SCT)

Anhedonia was assessed during the last week of uCMS using the SCT. Briefly, animals were habituated to the sucrose solution for 1 h during three trials the week before the uCMS protocol to establish baseline preference levels. To test sucrose preference, animals were food-and water-deprived for 12 h and were then presented with two pre-weighed bottles containing 2% sucrose solution or tap water for a period of 1 h (starting at the beginning of the dark period). Sucrose preference was calculated as previously described [36]. 

#### 2.2.2. Novelty Suppressed Feeding (NSF)

Anxiety-like behavior was assessed through the NSF test at the end of the uCMS protocol. Animals were food-deprived for 18 h and were then placed in an open-field arena for a maximum of 10 min, where a single food pellet was positioned in the center, as previously described [36]. After reaching the pellet, the animals were returned to their home cage to feed for 10 min. The latency to feed in the open-field arena was used as an anxiety-like behavior index; food consumption in the home cage provided a measure of appetite drive.

#### 2.2.3. Forced Swimming Test (FST)

Depressive-like behavior was assessed at the end of the uCMS protocol using the FST. Test trials were conducted 24 h after a 5 min pretest session. For that, rats were placed in glass cylinders filled with water (23 °C; 50 cm deep) for 5 min. An increase in immobility time was taken as a measure of depressive-like behavior. 

### 2.3. Corticosterone Levels Measurement

Corticosterone levels were measured in blood serum using a [^125^I] radioimmunoassay (RIA) kit (ICN Biochemicals, Costa Mesa, CA, USA) according to manufacturer’s instructions. The sensitivity of the RIA was 12 ng/mL (coefficient of variation of approximately 5%, inter-assay variation of approximately 7%, and intra-assay variability <5%, based on >150 assays).

Blood sampling (tail venipuncture) was performed during the diurnal nadir (N, 08:00–09:00) at the sixth week of the uCMS protocol. Blood was collected in non-heparinized polypropylene tubes, allowed to clot at room temperature, and centrifuged at 2500 g to recover serum that was then stored at −20 °C until further analysis. All animals included in this study (*n* = 10) were assessed for corticosterone levels.

### 2.4. BrdU Administration

BrdU is a thymidine analog that can be incorporated into newly synthesized DNA during the S-phase of the cell cycle. To assess proliferation and cell cycle exit, BrdU (100 mg/kg in 0.9% saline solution; Sigma-Aldrich, St. Louis, MI, EUA) was administered to the animals 24 h before sacrifice by intraperitoneal injection. 

### 2.5. Neurospheres Culture

Postnatal day 5 Wistar Han rats (*n* = 10) were rapidly decapitated using scissors, and their brains removed. The meninges were removed, the hemispheres were separated, and the hippocampus was macrodissected in ice cold DMEM-FBS 10%. After mechanical trituration and washes in DMEM-FBS 10%, hippocampal cells were seeded in 12-well plates (NUNC) in 3 mL of neurosphere medium DMEM-F12-GlutaMAX™, B27 2%, Pen-Strep 1%, HEPES buffer 8 mM, and bFGF and EGF (10 ng/μL). Cells were maintained at 37 °C in 5% CO_2_ and a humid atmosphere. Every 2 days, dexamethasone (DEX, Fortecortin, Merck; 1 μM) was added to the medium. In the last 4 days of culturing, 5-HT (Serotonin hydrochloride, Sigma) was added to the culture. Neurospheres were harvested at day 7 for further analysis.

For the differentiation experiments, the neurospheres were split at day 7 using Accutase Cell Detachment Solution (Merck Millipore). Viable cells were counted by means of trypan blue exclusion assay in a hemocytometer and plated in Poly-D-Lysine-coated 24-well plates at a density of 30,000 cells/well. Cells were maintained at 37 °C in 5% CO_2_ and in a humid atmosphere for 7 days.

### 2.6. Immunostaining Procedures

#### 2.6.1. *In Vivo*

Animals (*n* = 5/group) were deeply anesthetized with sodium pentobarbital (100 mg/kg; 20%; Eutasil, Sanofi) and were transcardially perfused with 0.9% saline solution followed by cold 4% paraformaldehyde. Brains were removed and post-fixed in 4% paraformaldehyde. Twenty-four hours later, the brains were washed with phosphate-buffered saline (PBS) solution, embedded in OCT compound (Thermo Scientific, Waltham, MA, EUA), and frozen at −20 °C. Coronal cryosections (20 µm) were double-stained for BrdU (mouse monoclonal; 1:50; Dako, Glostrup, Denmark) and Ki67 (rabbit polyclonal; 1:300; Merck Millipore, Darmstadt, Germany) or for BrdU and Doublecortin (rabbit polyclonal, 1:200, Merck Millipore, Darmstadt, Germany). Briefly, tissue sections were permeabilized in 1× PBS/Triton-X 100 0.5% for 10 min followed by washes with 1× PBS. Antigen-retrieval was performed with sodium citrate buffer for 15 min in the microwave. Sections were then washed in 1× PBS and were acidified using HCl for 30 min prior to overnight incubation with primary antibodies (1× PBS, 10% Fetal Bovine Serum, FBS) at 4 °C. After the primary antibody step, sections were washed in 1× PBS followed by secondary antibody incubation (1:1000; anti-mouse Alexa-fluor^®^ 488; anti-mouse Alexa-fluor^®^ 594; anti-rabbit Alexa-fluor^®^ 488; anti-rabbit Alexa-fluor^®^ 594; Life Technologies, Thermo Fisher Scientific; MA, USA, Appendix A) and were incubated for 2 h at RT (antibodies on Appendix A). Finally, all sections were stained with 4’,6-diamidino-2-phenylindole (DAPI; 1 mg/mL). For each animal, BrdU positive, Ki67 positive, and double positive cells within the SGZ of the hDG were counted (6–8 brain sections per animal) using confocal microscopy (Olympus FluoViewTM FV1000, Hamburg, Germany). The corresponding DG areas were determined using a motorized microscope (Axioplan 2; Carl Zeiss, LLC, White Plains, NY, USA) and the Neurolucida software (MBF Bioscience, Williston, VT) to estimate cell densities. Cell cycle exit was determined as the number of BrdU+ Ki67- cells per total BrdU+ cells. Cell counts from each animal were averaged. Results are presented as the average of total cell counts in each experimental group. 

#### 2.6.2. *In Vitro*


For neurosphere characterization, cells were plated after 7 DIV in Poly-D-Lysine (PDL)-coated slides for 2 h (neurospheres immunostaining). For differentiation capacity assessment, neurospheres were split (as previously described in Neurosphere culture section) and plated in PDL-coated slides at a density of 30,000 cells/well (24-well plates) for 7 days. For proliferation analysis, cells were split and plated in PDL-coated slides for 24 h with 1 μM BrdU (Sigma). Cultures were then fixed in 4% PFA for 10 min at RT and were then washed with PBS. PBS-T 0.5% was used to permeabilize cellular membranes for 10 min. Incubation with primary antibodies for Nestin (Millipore), GFAP (Dako), β-3 tubulin (Sigma), Ki67 (Millipore), O4 (R&D Systems), or BrdU (Dako) was performed overnight at 4 °C (antibodies on Appendix A). Primary antibodies were detected using subclass specific secondary antibodies (1:1000; anti-mouse Alexa-fluor^®^ 488; anti-rabbit Alexa-fluor^®^ 594; anti-mouse Alexa-fluor^®^ 647; anti-rat Alexa-fluor^®^ 594; Life Technologies, Thermo Fisher Scientific; MA, USA, Appendix A) and were incubated for 2 h at RT. Finally, a 10 min incubation with DAPI (Invitrogen, Waltham, MA, USA) was performed. Slides were then mounted with PermaFluor mounting medium (Thermo Scientific) and were observed under an Olympus BX-61 Fluorescence Microscope (Olympus, Düsseldorf, Germany). 

For proliferation analysis, BrdU positive, Ki67 positive, and BrdU/Ki67 double positive cells were counted. For this purpose, two/three coverslips and ten randomly selected microscope fields per condition were analyzed. Results are shown as the average number of BrdU+ or Ki67+ cells per DAPI.

### 2.7. Cell Cycle Distribution

To estimate the distribution of neural progenitor cells (NPCs) within the G1, S, and G2/M phases of the cell cycle, propidium iodide (PI; Invitrogen) staining was performed. PI is a fluorescent molecule (excitation max. =535 nm; emission max.=617 nm) and an intercalating agent that can be used to stain cells. PI is used to quantitatively assess DNA content as it binds stoichiometrically to DNA by intercalating between the bases and has little or no sequence preference. In this way, cells in the S phase (DNA synthesis phase) will take up proportionally more dye and will fluoresce more brightly than cells in G1; cells in G2 present doubled DNA content and thus will be approximately twice as bright as cells in G1.

Briefly, animals (*n* = 3/experimental group) were deeply anesthetized with sodium pentobarbital (100 mg/kg, 20%; Eutasil, Sanofi) and were transcardially perfused with 0.9% saline solution. Hippocampal DG (hDG) was macrodissected and dissociated in cold 1× PBS using a 70 µm cell strainer. The resulting cellular suspension was centrifuged, and cells were resuspended in 70% ethanol and were kept at −20 °C for at least 24 h, for fixation. 

For *in vitro* cultures, neurospheres (*n* = 5) were split using Accutase cell detachment solution for 10 min (Merck Millipore, Darmstadt, Germany) followed by mechanical dissociation with a micropipette (P200) and were then centrifuged for 5 min at 1000 rpm. After dissociation, single cells were resuspended in 70% ethanol and were kept at −20 °C for at least 24 h for fixation. 

After fixation, both the hDG cells and the *in vitro* neurosphere-derived NPCs were washed in PBS 1× and were then incubated with PI staining solution (PI in PBS/Triton-X 100 and RNAse A (20 mg/mL; Invitrogen) for 1 h at RT.

The cell cycle distribution analysis was performed using a FACS Diva flow cytometry system (BD Biosciences, San Jose, CA, USA). Cells were first gated on the forward/side scatter plot to eliminate debris and to reduce auto fluorescence prior to analysis. Then, singlets were gated according to DNA content represented by PI intensity (area vs. width). A histogram was used to determine events representing cells with distinct DNA content; double DNA content (double fluorescence intensity) corresponded to cells in G2/M phase compared to the G0/G1 phase, and cells with intermediate DNA content (intermediate fluorescence) corresponded to the S phase of the cell cycle. At least 10,000 PI-stained events were collected for each analysis.

### 2.8. Gene Expression Analyses

#### 2.8.1. RNA Purification

Animals were deeply anesthetized with sodium pentobarbital (20%; Eutasil, Sanofi) and were transcardially perfused with 0.9% saline solution. Brains were, removed and the hippocampal DG was macrodissected. Immediately after dissection, tissues were frozen and stored at −80°C until further analysis.

Total RNA was isolated from the macrodissected DG (*n* = 4–5/group) or from hippocampal neurospheres (*n* = 4/group) using the Direct-zol™ RNA MiniPrep (Zymo Research, CA, USA) according to the manufacturer’s instructions. RNA quantification and quality were assessed using a NanoDrop spectrophotometer. A260/230 and A260/280 ratios between 2.0 and 2.2 were accepted as good RNA quality. 

#### 2.8.2. cDNA Synthesis and Real-Time PCR Analysis

Total RNA (500 ng) was reverse transcribed using qScript™ cDNA SuperMix (Quanta Biosciences™, Gaithersburg, MD, USA). For real time RT-PCR, oligonucleotide primers for selected genes of interest were used (Appendix A). Reactions were performed in an Applied Biosystems 7500 Fast Real-Time PCR System (Applied Biosystems, LLC, Foster City, CA, USA) using 5x HOT FIREPol^®^ EvaGreen^®^ qPCR Mix Plus, ROX (Solis Biodyne, Tartu, Estonia). Target gene expression levels were normalized against the housekeeping gene Beta-2-Microglobulin (B2M). The relative expression was calculated using the ΔΔCt method. Results are presented as mean relative gene expression levels after normalization to B2M mRNA levels and to CT samples.

### 2.9. Statistical analysis

Statistical analysis was performed using the GraphPad Prism 6.0 software (GraphPad Software, Inc., La Jolla, CA, USA). The underlying assumptions for all of the statistical procedures were assessed. The normal distribution was tested using the Kolmogorov–Smirnov test. Student’s t-test was used to assess differences between the control and uCMS groups (to test for the effect of uCMS) and between the uCMS and uCMS + FLX groups (to test for the effect of antidepressant treatment). Pearson correlation was used to assess correlation between the number of BrdU+ and Ki67+ cells in the DG and the percentage of cells exiting the cell cycle. Data transformations were tested when the described assumptions were violated. As these transformations did not prove to be useful to accomplish normality or homogeneity of variances, the alternative non-parametric tests were applied (Mann–Whitney test; results can be found on Appendix A). As all of the significant results remained the same, the results for the parametric statistical tests are presented. Test statistics and *p*-values are shown for each test. Significance was set at *p* < 0.05.

## 3. Results

### 3.1. In Vivo Experiments

#### 3.1.1. Behavioral Characterization of the Animal Model of Depression

The animal model of depression used in this study, the unpredictable chronic mild stress model (uCMS; Appendix A), uses stress as an etiological factor for the development of depressive and anxiety-like behavior in rodents. Here, we exposed animals to chronic mild stressors for 6 weeks. During the last 2 weeks of this protocol, fluoxetine, a commonly prescribed selective serotonin reuptake inhibitor (SSRI), was administered to the animals (Figure 1A). Behavioral analysis shows that uCMS-exposed animals presented an anhedonic profile, as demonstrated by the decreased levels of sucrose preference in the sucrose consumption test (SCT) (Figure 1B; SCT: CT = 94.10 ± 1.245% vs. uCMS = 87.79 ± 1.673%, t_18_ = 3.029, *p* = 0.0072). Moreover, the uCMS animals presented increased latency to feed in the novelty suppressed feeding (NSF) test, revealing an anxiety-like behavior (Figure 1C; CT = 95.38 ± 25.91 s vs. uCMS = 273.8 ± 25.23 s, t_18_ = 4.934, *p* = 0.0001) and depressive-like behavior in the forced swimming test (FST) (Figure 1D; CT = 143.3 ± 31.81 s vs. uCMS = 246.0 ± 15.31 s, t_18_ = 2.909, *p* = 0.0094). To validate significant changes in latency to feed assessed by the NSF, the alterations in appetite drive must be characterized. Although uCMS-exposed animals presented significantly higher appetite drive when compared to the control (CT) animals, they also showed an increased latency to feed, thus validating the NSF results (Appendix A). This effect may be due to the fact that during the uCMS protocol, though the total food intake throughout a 24 h period was similar between the control and uCMS rats, they presented a distinct pattern of food intake during light hours compared to dark hours [37]. Because the control animals fed more during dark hours, which corresponds to their “naturally” active period, whereas the uCMS-exposed animals fed both during the dark as well as during the light hours, the uCMS animals may have been more prone to ingest an increased amount of food during the light hours, which is when the NSF test was performed.

Chronic fluoxetine (FLX) treatment was able to reverse these behavioral deficits (Figure 1B, SCT: uCMS + FLX = 94.69 ± 1.615% vs. uCMS = 87.79 ± 1.673%, t_18_ = 2.971, *p* = 0.0082; Figure 1C, NSF: uCMS + FLX =141.4 ± 33.81 s vs. uCMS = 273.8 ± 25.23 s, t_18_ = 3.140, *p* = 0.0057; Figure 1D, FST: uCMS + FLX =144.6 ± 28.5 s vs. uCMS = 246.0 ± 15.31 s, t_18_ = 3.132, *p* = 0.0058) to the levels of the CT animals. Additionally, corticosterone level assessment showed a significant increase in uCMS animals (Figure 1E, CT = 76.05 ± 15.10 ng/mL vs. uCMS = 153 ± 28.04 ng/mL, t_18_ = 2.417, *p* = 0.0260) that was restored by fluoxetine administration (Figure 1E, uCMS + FLX = 85.69 ± 7.223 ng/mL vs. uCMS = 153 ± 28.04 ng/mL, t_18_ = 2.336, *p* = 0.0313).

#### 3.1.2. Proliferation and Generation of Newborn Neurons

A reduced number of proliferating cells in the hDG was observed in the uCMS-exposed animals, as assessed using both the endogenous marker Ki67 (Figure 1F; CT = 24.59 ± 3.325 vs. uCMS = 15.11 ± 1.683, t_8_ = 2.686, *p* = 0.0250) and the exogenous marker BrdU (Figure 1F; CT = 14.16 ± 0.9141 vs. uCMS = 7.643 ± 0.7548, t_8_= 5.494, *p* = 0.0006), and fluoxetine treatment was able to reverse these numbers to the levels of the CT animals (Figure 1F; Ki67: uCMS + FLX = 27.41 ± 5.306 vs. uCMS = 15.11 ± 1.683, t_8_= 2.621, *p* = 0.0306; BrdU: uCMS + FLX = 20.14 ± 2.102 vs. uCMS = 15.11 ± 1.683, t_8_ = 5.595, *p* = 0.0005). Moreover, the number of newly born neurons, which was assessed by the number of BrdU/DCX double-positive cells, was decreased in the uCMS-exposed animals (Figure 1G; CT = 14.94 ± 3.124 vs. uCMS = 6.422 ± 0.9598, t_8_ = 2.606, *p* = 0.0313) and was restored by chronic fluoxetine treatment (Figure 1G, uCMS + FLX = 11.15 ± 1.068 vs. uCMS = 6.422 ± 0.9598, t_8_= 3.292, *p* = 0.0110). 

#### 3.1.3. Cell Cycle Kinetics Analysis in the uCMS Animal Model of Depression

Cell cycle distribution analysis using PI staining, revealed that the uCMS-exposed animals presented a slightly increased proportion of hDG cells in the G0/G1 phase of the cell cycle and a significantly decreased proportion in the G2/M phases (Figure 1H; G0/G1: CT = 77.55 ± 1.050 vs. uCMS = 80.15 ± 0.5500, t_4_ = 2.193, *p* = 0.0933; G2/M: CT = 16.95 ± 0.850 vs. uCMS = 14.40 ± 0.300, t_4_ = 2.829, *p* = 0.0474). Animals treated with fluoxetine presented significantly decreased levels of cells in the G0/G1 phase of the cell cycle when compared to the untreated uCMS animals (Figure 1H; uCMS + FLX = 76.40 ± 0.800 vs. uCMS = 80.15 ± 0.5500, t_4_ = 3.863, *p* = 0.0181), but no differences in the G2/M phases were found (Figure 1H; uCMS + FLX = 16.45 ± 2.750 vs. uCMS = 14.40 ± 0.300, t_4_ = 0.7411, *p* = 0.4998). Additionally, the proportion of cells exiting the cell cycle was evaluated (Figure 1I). This ratio provides information about the cells that have entered the cell cycle and that have incorporated BrdU during the S phase (BrdU+ cells) 24 h before sacrifice at the time of BrdU injection but that were no longer cycling when the animals were sacrificed (Ki67- cells), thus meaning that they had exited the cell cycle. Although not statistically significant, the uCMS-exposed animals presented a slightly decreased proportion of cells exiting the cell cycle compared to the CT animals (Figure 1I; CT = 33.25 ± 3.076 vs. uCMS = 18.43 ± 6.706, t_8_ = 1.872, *p* = 0.0940). Interestingly, fluoxetine could increase these levels when compared to the untreated uCMS animals, although this was not statistically significant (Figure 1I; uCMS + FLX = 37.5 ± 3.182 vs. uCMS = 18.43 ± 6.706, t_8_ = 2.179, *p* = 0.0610). To test if the proportion of cells exiting the cell cycle could depend on the number of proliferating cells, we performed a correlation analysis between the proportion of cells exiting the cell cycle and the number of BrdU+ and Ki67+ cells. As expected, there was a significant correlation between the levels of BrdU+ and Ki67+ cells in the hDG (Pearson r: 0.6573, R^2^ = 0.4320, *p* = 0.0078). Importantly, proliferation levels (assessed using both BrdU or KI67) were not correlated with the proportion of cells exiting the cell cycle (BrdU: Pearson r: 0.3561, R^2^ = 0.1268, *p* = 0.1927; Ki67: Pearson r: 0.3719, R^2^ = 0.1383, *p* = 0.1723), thus emerging as an independent measure of the cell cycle dynamics (Appendix A). 

#### 3.1.4. Analysis of the Cell Cycle Regulators Expression *In Vivo*

To unveil the cell cycle molecules that could be responsible for the changes observed in the cell cycle of the hippocampal progenitor cells from the uCMS-exposed animals and animals treated with fluoxetine, we analyzed the expression of cyclins and Cdk inhibitors involved in the G1 and S-phases of the cell cycle. qRT-PCR analysis in the hDG of uCMS animals revealed decreased expression levels of two G1 cyclins, cyclin D1 and cyclin E (Figure 1J; cyclin D1: t_7_ = 3.517, *p* = 0.0098; cyclin E: t_7_ = 11.35 *p* < 0.0001), and cyclin A, an S-phase and S/G2 transition cyclin (Figure 1J; cyclin A: t_8_= 2.049 *p* = 0.0373). Fluoxetine treatment was only able to significantly restore the decreased levels of cyclin E to those of the CT animals (Figure 1J; cyclin E: t_8_= 4.691 *p* = 0.0016). No significant differences were observed in the expression of Cdks or Cdk inhibitors (Figure 1K,L).

### 3.2. In Vitro Experiments

To further understand the molecular regulation of hippocampal progenitor cell proliferation after stress and AD treatment, we used an *in vitro* approach—the neurosphere culture. This approach allowed us to further dissect the possible mechanisms responsible for the changes observed in our *in vivo* analysis using an enriched population of proliferative cells. 

#### 3.2.1. Neural Progenitor Cells Proliferation and Differentiation

After 7 days *in vitro* (DIV) isolated NPCs grow in sphere-like structures when stimulated with the growth factors EGF and bFGF (Figure 2A,B). To assess their ability to differentiate in different neural cell types, neurospheres were split and plated in PDL-coated slides for an additional 7 day-period. Immunostaining analyses showed that they could differentiate between neurons, astrocytes, and oligodendrocytes (Figure 2B), thus making this a suitable tool to study proliferation and differentiation phenomena occurring in the hippocampal neurogenic niche.

#### 3.2.2. Proliferation and Cell Cycle Regulation in DEX-Stimulated Neurospheres and After 5-HT Treatment

To mimic the effects of the elevated GC levels observed in uCMS-exposed rats (Figure 2A), we stimulated the NPCs with DEX, a GR agonist. NPCs were grown in the presence of 1 μM DEX for 7 days. DEX was added to the culture every second day. During the last 4 days of culturing, part of the cells was additionally stimulated with 1 μM of 5-HT to mimic the pro-serotonergic effects of SSRIs treatment, which was also observed in a preliminary analysis of depressive-like animals treated with fluoxetine (data not shown).

DEX treatment for 7 days promoted decreased proliferation of NPCs, as assessed by BrdU immunostaining (Figure 2C; CT = 0.17 ± 0.02 vs. DEX = 0.05 ± 0.02, t_4_ = 4.243, *p* = 0.0132). Although the number of Ki67+ cells was similar between DEX-treated and untreated cells (Figure 2D), the number of BrdU+ Ki67+ double-positive cells was decreased in the DEX-treated NPCs (Figure 2E; CT = 0.1200 ± 0.0100 vs. DEX = 0.0250 ± 0.0150, t_4_ = 5.270, *p* = 0.0062). The 5-HT treatment was able to significantly reverse the number of BrdU + Ki67+ double-positive cells (Figure 2E; DEX + 5-HT = 0.08 ± 0.01528 vs. DEX = 0.0250 ± 0.0150, (one-tailed t test) t_3_ = 2.426, *p* = 0.045).

Cell cycle distribution, as assessed by PI staining, revealed a G1 phase arrest in the DEX-treated cells, which was shown by a significantly increased percentage of cells in the G0/G1 phase and a decreased percentage of cells in the S phase (Figure 2F; G0/G1: CT = 78.55 ± 2.100 vs. DEX = 84.72 ± 1.230, t_8_ = 2.535, *p* = 0.0350; S: CT = 18.220 ± 1.100 vs. DEX = 10.10 ± 0.200, t_8_= 7.263, *p* < 0.0001) that was only partly and not significantly rescued by 5-HT treatment (Figure 2F; G0/G1: DEX = 84.72 ± 1.230 vs. DEX + 5-HT = 81.93 ± 1.200, t_8_= 1.624, *p* = 0.1431; S: DEX = 10.10 ± 0.200 vs. DEX + 5-HT = 11.17 ± 1.400, t_8_= 0.7566, *p* = 0.4710). Moreover, preliminary data from our lab (not shown) suggest that DEX treatment promotes decreased metabolic viability and increased cell death.

To elucidate the modest effect of 5-HT in reversing the DEX-induced phenotype, we analyzed the expression of the GR and 5-HT receptors that were previously reported to be involved in the proliferation of NPCs in the hippocampal neurogenic niche and in the action of ADs [26,27]. These NPCs express GR and the 5-HT receptors 5-HT1a and 5-HT2a (Appendix A). Notably, 5-HT treatment could partly increase the expression of GR and significantly increase the expression of 5-HT2a when compared to DEX-treated cells (t_4_ = 11.37; *p* = 0.0003; Appendix A).

#### 3.2.3. Cyclins and CDK Inhibitors Expression after *In Vitro* Stimulation with DEX and 5-HT

As DEX was able to promote an arrest in the cell cycle progression we sought to explore the expression of cyclins and Cdk inhibitors in these cells. DEX treatment significantly decreased the expression of cyclin D1 (Figure 2G; t_6_ = 2.801, *p* = 0.0311) and cyclin D2 (Figure 2G; t_6_ = 2.565, *p* = 0.0426). Interestingly, and despite the modest effect on reversing the DEX-induced changes in proliferation and cell cycle distribution, 5-HT treatment restored the levels of cyclin D1 to those of the CT cells (Figure 2G; t_6_ = 3.490, *p* = 0.0130) but only partly to those of cyclin D2 (Figure 2G). Additionally, the levels of Cdk6 were significantly decreased in Dex-treated cells (Figure 1H, t_6_ = 2.518, *p* = 0.0454) and reversed were by fluoxetine treatment (Figure 1H, t_6_ = 3.158, *p* = 0.0196). Moreover, the levels of p27, a Cdk inhibitor, were significantly increased in the DEX-exposed cells (Figure 2I; t_6_ = 2.727, *p* = 0.0339) and were not reversed by 5-HT treatment (Figure 2I).

## 4. Discussion

Many studies by our group and others have supported a role for adult hippocampal neurogenesis in the onset and remission from depressive-like behaviors [15,20,22,38,39,40,41,42,43,44,45,46,47,48]. Still, the molecular changes, namely at the cell cycle level, that are responsible for the regulation of these proliferative and neurogenic phenomena are largely unknown. Understanding the factors regulating the generation of new neurons in the adult hDG may open new avenues for the development of more targeted therapeutic strategies to restore hippocampal cell genesis and to improve remission.

A major hurdle when studying the processes controlling the cell cycle in the adult hippocampus is the small proportion of proliferating cells at any given time. The adult hDG is mainly composed of post-mitotic cells, thus limiting the ability to dissect the cell cycle phenomena. To somehow overcome this limitation, we used a multidimensional approach. Considering the relevance of maintaining the niche environment, we used an *in vivo* approach by studying the hDG of an uCMS rat model of depression. Additionally, we used an *in vitro* system—the neurosphere culture—to study the hippocampal-derived proliferating cells, specifically (Figure 3). This approach allowed us to explore this question from two distinct but complementary perspectives and allowed us to take advantage of each of the models. By using DEX and 5-HT we mimicked, in a simplistic way, the stress and AD treatment conditions; increased GC levels are observed after stress and in many cases of depression [49,50,51,52] and were also confirmed in our model. Additionally, preliminary data from our uCMS model show an increase in 5-HT levels in the hippocampus after the chronic administration of fluoxetine, as described in previous studies using SSRIs [51].

The uCMS animal model of depression recapitulates many of the core behavioral and neuroplastic changes that are characteristic of depression and is responsive to antidepressant therapy [15,35,52,53,54]. It is thus a suitable tool to study the relationship between chronic stress, a potent precipitating factor for depression, and the molecular, cellular, and behavioral changes that occur in a depressive state. Here, and as previously reported [20,55], we observed a decrease in proliferation (decreased expression of BrdU and Ki67) and in the number of newly born neurons in the hDG of uCMS-exposed animals, which was reversed by chronic treatment with fluoxetine. Cell cycle analysis data suggested that the decreased number of proliferating cells in uCMS-exposed animals may result from a delayed progression or cell cycle arrest in the G1 phase that can be overpassed by fluoxetine treatment. Firstly, the reduced numbers of Ki67+ cells may suggest that stem cells are less recruited to enter the cell cycle, which is probably due to decreased proliferative cues [24] and available neurotrophic factors [56]. Moreover, if cells enter the G1 phase of the cell cycle they are less likely to go through S, G2, and M phases and give rise to a newborn cell, as implied by the decreased numbers of BrdU+ positive cells and the decreased proportion of cells in G2/M phase. In support of this interpretation, there was an indication of a slight decrease in the proportion of cells exiting the cell cycle in uCMS animals. Still, further analyses are needed to confirm this possibility, and assessing the cell cycle length of these cells could help to elucidate the phenomenon.

These changes in cell cycle kinetics were accompanied by a decreased gene expression of the G1 and S/G2 phase transition cyclins, namely cyclin D1, E, and A (Figure 3). Because these cyclins can also be expressed in post-mitotic cells, where they have alternative functions (other than those related to the cell cycle regulation) [57,58], this data must be carefully interpreted. Previous reports have shown that GCs decrease embryonic stem cells proliferation by promoting the ubiquitin-mediated degradation of cyclin D1 [59], suggesting an additional GC-mediated post transcriptional regulatory mechanism for decreasing cyclin D1 expression. Evaluating the protein expression of these cyclins using co-labelling methods will help disclose their implications in the regulation of the cell cycle in progenitor cells.

The results from the *in vitro* approach support the cell cycle arrest hypothesis, with DEX treatment having similar anti-proliferative and G1 arrest effects on hNPCs. No changes were detected in the number of Ki67+ cells upon DEX treatment, which may be in line with a G1 phase arrest hypothesis, as previous studies have shown that Ki67 may also be expressed in G1 arrested cells [60]. As previously described for several models [61,62,63], preliminary data from our lab further suggest a role for DEX in promoting the cell death of hNPCs, which may occur as a consequence of the G1 phase arrest. Consistent with this hypothesis, previous findings in T-lymphoma cells show that DEX-mediated cell cycle arrest is followed by apoptosis [64]. Future work may disclose whether cell death also occurs *in vivo* and the putative role of antidepressants in preventing this effect.

As observed in the hDG of uCMS-exposed animals, DEX treatment decreased the gene expression of cyclin D1 in proliferating cells *in vitro*. Additionally, it decreased the levels of cyclin D2 and Cdk6 and increased p27 expression (Figure 3). GCs have been consistently reported as regulators of proliferation [32,65,66]. GRs are expressed in most neuronal developmental stages, including in stem cells and transit amplifying progenitors (tAPs) but not in committed neuronal progenitors [67]. In line with this, our findings suggest that the regulation of proliferation by stress most likely occurs in proliferating stem cells or tAPs at the level of the G1 phase and that these changes may be in part mediated by GCs. GCs classically exert their action on gene expression through the activation of the cytoplasmatic glucocorticoid receptors (GRs) that bind to the glucocorticoid response elements (GREs). Using the FIMO in silico tool [68], we found the previously identified GRE sequence 5’-TGTCCT-3’ [64,69,70] in the rat cyclin D1 promoter at position -389 relative to the transcription start site (Appendix A). More investigation is needed to assess whether this is a functional GRE.

In our experiments, 5-HT could not completely reverse the DEX-induced proliferation reduction, which may suggest the need for additional effector molecules or more complex cellular interactions for the pro-proliferative effects to occur. Other factors may be required to trigger paracrine and/or downstream effects, such as an increase in the expression of neurotrophic factors or the activation of signaling pathways. Though we have used standard 5-HT concentrations, this lack of effect could also have been due to low 5-HT concentration or treatment duration. Alternatively, there is a chance that this *in vitro* model is overly simplistic when compared to the hippocampal neurogenic niche. A previous study, using a reporter mouse line, showed that chronic fluoxetine treatment specifically targets transit amplifying neural progenitors (tANPs), increasing the number of symmetric divisions of these cells [71]. Though the cell cycle mechanisms responsible for that observation were not explored, their data endorse the validity of our *in vitro* approach, which is representative of the proliferating progenitor cells in the hDG. In that study, however, fluoxetine-treated animals had not been previously challenged with any stressor, and the mechanisms regulating these cells in a depressive-like condition may be different.

Most of the studies trying to disclose cell cycle dynamics have been performed in constitutive KO animal models [72,73,74,75]. Some have shown that the targeted deletion of cell cycle regulators does not yield any particular defects in hippocampal neurogenesis, which is most likely due to molecular redundancies and compensatory mechanisms occurring during development [76]. Such approaches may have underappreciated the relevance of these molecules in the fine-tuning of hNPCs proliferation.

Together, our results are in line with previous suggestions of a G1 phase arrest of the cell cycle after chronic stress [27]; in a previous report, Heine et al. disclosed an increased number of p27kip1+ cells in the hippocampal SGZ after exposure to an unpredictable stress paradigm for 3 weeks [27]. In our *in vitro* experiments, we were also able to disclose the increased expression of this Cdk inhibitor after DEX treatment. Even though distinct methodologies were used, as Heine et al. used stereological protein quantification analyses of p27kip1+ [27], these results are in agreement and suggest a GC-mediated cell cycle arrest that can be promoted by increased p27 expression and the decreased expression of G1 cyclins. Moreover, regarding the possibility of senescence induction, we could not detect significant differences in the expression of one of the most well-established senescence markers p21, a Cdk inhibitor downstream of phospho-p53, which has also been described to be involved in the pro-neurogenic effects of antidepressants in unstressed rodents [77].

Future studies should focus on specific cell types and use other complementary models to definitively disclose the underpinnings of cell cycle regulation in the context of depression and antidepressant action. Moreover, the more recently described differential impact of stress and antidepressants in the neurogenic process along the septo-temporal and transversal axes of the hDG [55,78,79] urges the need to explore putative differential cell cycle control mechanisms in the dorsal versus ventral and in the supra versus infrapyramidal blades of the hDG. Additionally, the effects of different classes of antidepressants in cell cycle regulation should be a matter of study to understand its implications for the treatment of depressed patients.

## Figures and Tables

**Figure 1 ijms-22-11798-f001:**
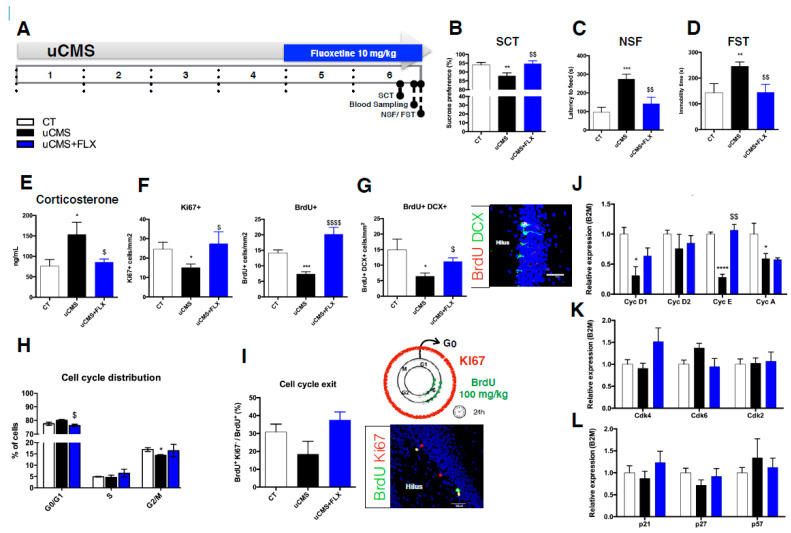
Cell cycle analysis in a uCMS animal model of depression. (**A**) Schematic representation of the experimental timeline used for the *in vivo* experiments. (**B**–**D)** Behavioral characterization of the animal model of depression. uCMS exposure induced an anhedonic profile (**B**) as well as anxiety (**C**) and depressive-like behavior (**D**). These changes were reversed by treatment with fluoxetine. (**E**). Diurnal nadir corticosterone levels measured in the serum of the animals. (**F**) Ki67 and BrdU (injected 24 h before sacrifice) and immunostaining cell densities in the hippocampal DG. (**G**). Number of BrdU/DCX double-positive cells (cell densities) in the hDG and BrdU/DCX immunofluorescence (scale bar = 40 μm). (**H**) Cell cycle distribution of hippocampal cells using PI staining and flow cytometry. (**I**). Cell cycle exit analysis using BrdU and KI67 immunostaining. Schematic representation of the BrdU administration paradigm and representative picture of the BrdU/Ki67 immunofluorescence in the DG. Scale bar = 50 μm. (**J**–**L**). Relative expression levels of G1 phase and S/G2 transition cyclins (**J**), CdKs (**K**), and Cdk inhibitors (**L**) in the hDG. Error bars denote SEM. * Denotes the effect of uCMS-exposure; ^$^ denotes the effect of fluoxetine compared with untreated uCMS-exposed animals. *^/$^
*p* < 0.05; **^/$$^ *p* < 0.01; *** *p* < 0.001; ****^/$$$$^ *p* < 0.0001. Abbreviations: CT: control; uCMS: unpredictable chronic mild stress; FLX: fluoxetine.

**Figure 2 ijms-22-11798-f002:**
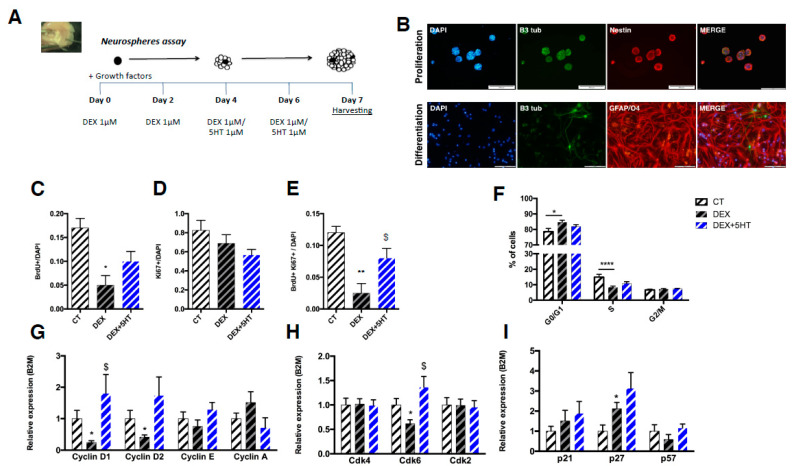
Neurosphere culture as a model to study the hippocampal progenitor cells proliferation and cell cycle. (**A**). Schematic representation of the neurosphere assay. (**B**). Representative pictures of immunofluorescence of neurosphere expansion and differentiation into neural phenotypes (scale bar = 100 μm). (**C**–**E**). Proliferation assessed by BrdU (24 h, **C**), Ki67 (**D**) and BrdU/Ki67 (**E**) immunostaining in split neurospheres. (**F**). Cell cycle distribution analysis with PI staining. (**G**–**I**). Gene expression analysis of G1 cyclins (**G**), Cdks (**H**), and CdK inhibitors (**H**). Error bars denote SEM. * Denotes the effect of DEX-stimulation; ^$^ denotes the effect of 5-HT compared DEX-treated cells. *^/$^ *p* < 0.05; ** *p* < 0.01; **** *p* < 0.0001. Abbreviations: CT: control; DEX – dexamethasone; 5-HT-serotonin.

**Figure 3 ijms-22-11798-f003:**
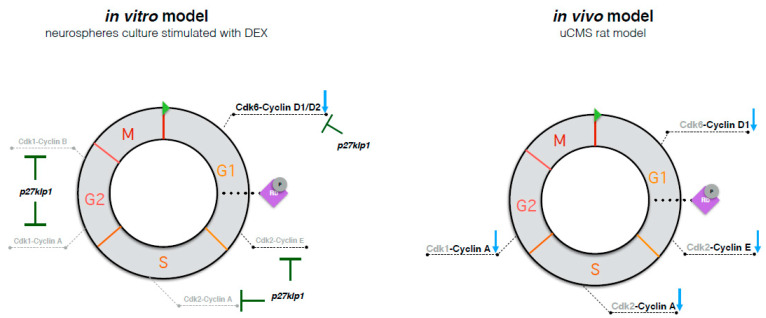
Schematic representation of the effects observed in cell cycle-regulatory molecules in both *in vitro* and *in vivo* experimental models used in this study.

## Data Availability

Data is contained within the article or Appendix A.

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
