# Peer review of "Cell Cycle Regulation of Hippocampal Progenitor Cells in Experimental Models of Depression and after Treatment with Fluoxetine"

_ijms, 2021, doi:10.3390/ijms222111798_

Round 1

Reviewer 1 Report

The manuscript “Cell cycle regulation of hippocampal progenitor cells in experimental models of depression and after treatment with fluoxetine” by Patricio et al studied tthe molecular mechanisms of the modulation of neurogenesis induced by chronic stress and modulated by antidepressants by investigating a chronic mild stress model in rats and in neurosphere cultures.

The results are interesting and the manuscript is clear.

There are some issues to be addressed by the authors:

  1. Statistical analysis: since three groups were compared, the most appropriate test was ANOVA, followed by post-hoc tests, instead of Student’s t test (line 416).
  2. Statistical analysis: when normality could not be reached by data transformation and non-parametric analyses were performed, those results should be reported, even though the results of the parametric tests were similar (line 422).
  3. The overall number of animals should be reported both in section 4.1 for in vivo experiments and in section 4.5 for neurosphere preparation.
  4. Since Supplementary materials are included to the manuscript, adding there a table in which the uCMS model stresses and time of administration were described would be useful to readers.
  5. Serum preparation method should be reported, as well as sacrifice method for gene expression studies and catalogue numbers of all antibodies.
  6. Figure S3 makes me wonder whether Dex-induced decrease in 5HT2a levels was statistically significant?
  7. Line 87 (and supplementary): the reason why uCMS induced increased appetite drive is not produced (or suggested).
  8. “Plated” is misspelled.

Author Response

The manuscript “Cell cycle regulation of hippocampal progenitor cells in experimental models of depression and after treatment with fluoxetine” by Patricio et al studied the molecular mechanisms of the modulation of neurogenesis induced by chronic stress and modulated by antidepressants by investigating a chronic mild stress model in rats and in neurosphere cultures.

The results are interesting and the manuscript is clear.

There are some issues to be addressed by the authors:

1) Statistical analysis: since three groups were compared, the most appropriate test was ANOVA, followed by post-hoc tests, instead of Student’s t test (line 416).

We understand the reviewer’s point. The t-test was used in this analysis because although we are studying three groups there are two different factors being assessed depending on the comparison:

-When comparing CT vs CMS we are addressing the CMS effect, so stress is the factor under study

-When comparing CMS vs CMS+Flx we are addressing the effect of the antidepressant.

If we had a control group treated with fluoxetine, we should have performed a Two-way ANOVA analysis to study these 2 factors together: stress and fluoxetine/antidepressant. In this work, however, our main goal was to understand the changes induced by fluoxetine in a pathological context, by modeling the human depressive condition, and find cell cycle-related molecules and mechanisms that could be targeted by this antidepressant. Accordingly, we only studied animals displaying depression-like behavior.

2) Statistical analysis: when normality could not be reached by data transformation and non-parametric analyses were performed, those results should be reported, even though the results of the parametric tests were similar (line 422).

We understand the reviewer’s concern, although parametric tests can be more robust, even when data normality is not achieved, and that’s why we have chosen to report those. In any case, we have now added a new table to the supplementary materials (Sup. Table 3), reporting the non-parametric test statistics for those analyses in which the datasets under comparison did not follow a normal distribution.

3) The overall number of animals should be reported both in section 4.1 for in vivo experiments and in section 4.5 for neurosphere preparation.

Regarding in vivo experiments, behavioral analysis and cort levels measurements have been performed in all animals (n=10, as stated in P. 4 – section 2.1 animals and treatments); This has now been added to the manuscript in P.5 and P6.

For neurosphere preparation we have used 10 rat pups. This information has now been added on P. 6.

4) Since Supplementary materials are included to the manuscript, adding there a table in which the uCMS model stresses and time of administration were described would be useful to readers.

We thank the reviewer for this important suggestion and have now added this information in the supplementary material (Supp. Figure 1) and referenced it on P 4 and 10 of the manuscript.

5) Serum preparation method should be reported, as well as sacrifice method for gene expression studies and catalogue numbers of all antibodies.

We have reported the serum preparation and sacrifice methods on P. 6 and P. 9 of the revised manuscript, respectively.

Catalogue numbers of all antibodies have now been compiled in a table in Supplementary material (Sup. Table 1).

6) Figure S3 makes me wonder whether Dex-induced decrease in 5HT2a levels was statistically significant?

Using t-test, we could not find statistically significant differences between CT and Dex-treated neurospheres (t6=1.606; p=0.1595). We have made this clearer by adding a sentence in Supplementary Fig. 4.

7) Line 87 (and supplementary): the reason why uCMS induced increased appetite drive is not produced (or suggested).

Previous data from our lab (Morais et al, New insights on the interplay between psychopharmacology and neuroplasticity in psychiatric disorders, PhD Thesis) show that during the uCMS protocol, though the total food intake throughout a 24h period is similar between Controls and uCMS rats, they present a distinct pattern of food intake during light hours compared to dark hours. In fact, control animals feed more during dark hours, which corresponds to their “naturally” active period, whereas uCMS-exposed animals feed both during the dark as well as during the light hours. This may explain why uCMS animals are more prone to ingest more food during the light hours, when the NSF test was performed. We have now added this suggestion to the revised manuscript (P.11).

Please see the Figure showing these results in the attached document.

8) “Plated” is misspelled.

We have corrected this typographical error on P.13.

Reviewer 2 Report

Paper entitled: “Cell cycle regulation of hippocampal progenitor cells in 2 experimental models of depression and after treatment with 3 fluoxetine” (ijms-1419815). The subject of the study is very interesting and requires deepening the poorly understood cell cycle mechanisms regulating hippocampal cell proliferation and genesis in depression and after antidepressant treatment. The presented results are interesting, but the confirmation of the effect of antidepressants such as fluoxytin requires many additional studies in this direction. In my opinion, making such far-reaching conclusions is too bold at this stage of research in such advancement. I think the conclusions should be polished. In addition, there are some issues which should be addressed before publication in IJMS.

  • The abbreviations shown in the figures - SCT, NSF, FST - are not meaningful and should be expanded in the text where they are presented for the first time, and in the description of the figures. The layout of the work in IJMS is different than in some journals, hence probably the development of shortcuts in the methodology, which is later in the manuscript. Unfortunately, there are so many shortcuts in the text that it's hard to get hold of it
  •  
  • Please provide the number of used tests for all the methodology used.
  •  
  • 3 Corticosterone levels measurement L322 - The methodology should be accurately described, and should include nr of assay, assay sensitivity, intraassy and interassay coefients of variation for corticosterone

  • L344 and 383 - what was the dosage of pentobarbital

Author Response

- Paper entitled: “Cell cycle regulation of hippocampal progenitor cells in 2 experimental models of depression and after treatment with 3 fluoxetine” (ijms-1419815). The subject of the study is very interesting and requires deepening the poorly understood cell cycle mechanisms regulating hippocampal cell proliferation and genesis in depression and after antidepressant treatment. The presented results are interesting, but the confirmation of the effect of antidepressants such as fluoxytin requires many additional studies in this direction. In my opinion, making such far-reaching conclusions is too bold at this stage of research in such advancement. I think the conclusions should be polished. In addition, there are some issues which should be addressed before publication in IJMS.

We thank the reviewer for his/her comment and have now improved the wording in the discussion section of the manuscript (P. 16-18) in order to make the conclusions less bold.

- The abbreviations shown in the figures - SCT, NSF, FST - are not meaningful and should be expanded in the text where they are presented for the first time, and in the description of the figures. The layout of the work in IJMS is different than in some journals, hence probably the development of shortcuts in the methodology, which is later in the manuscript. Unfortunately, there are so many shortcuts in the text that it's hard to get hold of it

We acknowledge the reviewer’s point and have now added the description of the abbreviations on P. 10 and 11 of the results section.

- Please provide the number of used tests for all the methodology used.

  We have now added the number of tests/animals/replicas for each test (P. 5 and 6).

- 3 Corticosterone levels measurement L322 - The methodology should be accurately described, and should include nr of assay, assay sensitivity, intraassy and interassay coefients of variation for corticosterone

 We have now added the requested information to the methods section 2.3 Corticosterone levels measurement (P. 5 and 6).

L344 and 383 - what was the dosage of pentobarbital

The dosage of pentobarbital was 100mg/kg. This information has now been added to the methods section (P. 6 and 8).

Reviewer 3 Report

The authors present very nice study regarding the control of cell cycle progression in hippocampal progenitor cells in experimental model of depression and after treatment with fluoxetine. The study has been executed in in vitro and in vivo models and deserves to be published. Howevere, there are several major points whicg need to be solved before publication. 

1) the expression level of all cell cycle regulators needs to be controled with Western Blots

2) please provide all details regarding antibodies (catalogue numbers, RRIDs)

3) the authors should analyze also the levels of p21/p53/p16

4) did you control the possibility of induction of senescence?

5) °C not oC - throughout the text 

Author Response

The authors present very nice study regarding the control of cell cycle progression in hippocampal progenitor cells in experimental model of depression and after treatment with fluoxetine. The study has been executed in in vitro and in vivo models and deserves to be published. However, there are several major points which need to be solved before publication.

1) the expression level of all cell cycle regulators needs to be controlled with Western Blots

We acknowledge the reviewer’s suggestion. We were able to optimize for this revision the WB for Cyclin D1, as this was a molecule that showed alterations in expression both in vitro and in vivo in our study. This WB analysis shows that uCMS decreases Cyclin D1 nuclear expression and that fluoxetine treatment partially recovers it to control levels, as observed for gene expression in Figure 1J.

Please see the Figure showing these results in the attached document.

2) please provide all details regarding antibodies (catalogue numbers, RRIDs)

Thanks for the suggestion. We have now included a table with this information in Supplementary materials (Sup. Table 1).

3) the authors should analyze also the levels of p21/p53/p16

We thank the reviewer for this comment. We have assessed the gene expression levels of p21, that is a CDKI downstream of phospho-p53, and no differences were found among groups both in vitro and in vivo (please see Figure 1L and Figure 2I). We were not able to assess those of p53 and p16, though these are also important markers of a stable cell cycle arrest and possible senescence.

4) did you control the possibility of induction of senescence?

We thank the reviewer for raising this question. In fact, following this possibility of senescence induction we analyzed one of the most well-established senescence markers, p21, and found no differences of its expression both in vitro and in vivo. We added this information in the discussion of the revised manuscript on P. 18.

We further assessed cell cycle arrest, which is also one of the hallmarks of cellular senescence, using PI staining. Additionally, we have assessed metabolic viability and cell death, using an MTS-based metabolic viability test and a PI/Annexin V staining protocol in the in vitro model. These preliminary results point to a decreased metabolic viability and increased cell death upon dexamethasone exposure (mentioned it in P. 14, preliminary data, n=1, see below).

Please see the Figure showing these results in the attached document.

5) °C not oC - throughout the text

This has been corrected throughout the text.

Round 2

Reviewer 1 Report

The authors have addressed all concerns 

Reviewer 2 Report

I have no comments 

Reviewer 3 Report

The Authors have improved  the manuscript.